# Wake redirection at higher axial induction

Carlo Cossu

Laboratoire d'Hydrodynamique Énergetique et Environnement Atmosphèrique (LHEEA)
CNRS - Centrale Nantes, 1 rue de la Noë 44300 Nantes, France
**Correspondence:** Carlo Cossu (carlo.cossu@ec-nantes.fr)

**Abstract.** The energy produced by wind plants can be increased by mitigating the negative effects of turbine-wakes interactions. In this context, axial induction control and wake redirection control, obtained by intentionally yawing or tilting the rotor axis away from the mean wind direction, have been the subject of extensive research but only very few investigations have considered their combined effect. In this study we compute power gains that are obtained by operating tilted and yawed rotors at higher axial induction by means of large eddy simulations using the realistic native NREL 5-MW actuator disk model implemented in SOWFA. We show that for the considered two-rows wind-aligned array of wind turbines the power gains, of approximately 5%, obtained by standard wake redirection at optimal tilt or yaw angles and reference axial induction can be more than tripled, to above 15%, by operating the tilted or yawed turbines at higher axial induction. It is also shown that significant enhancements of the power gains are obtained even for moderate overinduction. These findings confirm the potential of overinductive wake redirection highlighted by previous investigations based on more simplified turbine models that neglected wake rotation effects. The results also complement previous research on dynamic overinductive yaw control by showing that it leads to large power gain enhancements also in the case where both the yaw and the overinduction controls are static hopefully easing the rapid testing and implementation of this combined control approach.

## 1 Introduction

In wind farms, wind turbines shadowed by the wakes of other upwind turbines experience a decrease of the mean available wind speed and an increase of turbulent flucutuations resulting in decreased extracted wind power and increased fatigue loads (see Stevens and Meneveau, 2017; Porté-Agel et al., 2019, for a review). In currently installed wind farms, however, each turbine is typically operated in "greedy" mode maximizing its own individual power production. As the greedy operation mode does not generally lead to the global optimal, where the energy production of the whole wind farm is maximized (see e.g. Steinbuch et al., 1988), a number of different approaches have been proposed where the collective control of all turbines is used to increase the power production of the whole wind farm by mitigating the negative effects of turbine-wake interactions (see Knudsen et al., 2015; Boersma et al., 2017, for a review). Among the many proposed approaches, two have received particular

attention: axial induction control and wake redirection control which can be static (the control is steady if the incoming wind conditions are) or dynamic (the control can be unsteady even for steady incoming wind conditions).

In axial induction control the induction factors of selected (usually upwind) turbines are steered away from the greedy operation mode in order to increase the power production of other (usually downwind) turbines. While static axial induction control has not demonstrated significant power gains in realistic settings (Knudsen et al., 2015; Annoni et al., 2016), dynamic axial induction control has shown promise for significant power gains (Goit and Meyers, 2015; Munters and Meyers, 2017). In wake redirection control the intentional misalignment of rotor axes from the wind direction is used to deflect turbine wakes in the horizontal or in the vertical direction by acting on yaw or tilt angles respectively with a documented increase of the global power produced by the wind farm (Dahlberg and Medici, 2003; Medici and Alfredsson, 2006; Jiménez et al., 2010; Fleming et al., 2014, 2015; Campagnolo et al., 2016; Howland et al., 2016; Bastankhah and Porté-Agel, 2016).

In two recent studies (Cossu, 2020a, b) we have shown that an appropriate combination of (static) tilt and (static) axial induction control results in a significant enhancement of the global power gains obtained in spanwise-periodic wind-turbine arrays. In these studies, for the considered three-rows turbine arrays, power gains were observed to be highly enhanced (up to a factor of 2 or 3) when the turbines with rotor tilted by the optimal angle ($\varphi \approx 30^o$) were operated at disk-based thrust coefficient $C'_T = 3$ higher than in the baseline case ($C'_T = 1.5$).

The results reported in these previous studies (Cossu, 2020a, b) were obtained with an actuator-disk model where wake-rotation and the radial distribution of actuator-disk forces were neglected and the turbines were assumed to operate at constant given $C'_T$. This highly idealized setting, used in many previous investigations (e.g. Calaf et al., 2010; Goit and Meyers, 2015; Munters and Meyers, 2017), has been instrumental in obtaining general results not depending on the specific turbine control law and blade design but calls for confirmation on more realistic turbine models. Hence, a first goal of this study is to determine the power gains that can be obtained with high-induction (overinductive) tilt control when realistic turbine models are used that take into due account blade-design, wake-rotation and the controller specificity. This goal is addressed in the first part of this study, by making use of SOWFA's (Churchfield et al., 2012) native actuator disk model for the NREL 5-MW turbine. In this implementation of the turbine model the radial dependence of the actuator disk force as well as wake rotation and $C'_T$ are computed from turbine blades properties by means of a blade-element approach and NREL 5-MW's five-region realistic controller (Jonkman et al., 2009) is used.

In the second part of the study we address the case of yaw control. Indeed, the increased power gains obtained by operating tilted turbines at higher thrust coefficients mostly result from the increase of wake deviations obtained without a penalization of the power production of the tilted turbine. Overinductive wake deflection could therefore be beneficial also in the case of yaw-control where it is known that higher thrust coefficients also result in larger wake deviations (Jiménez et al., 2010; Howland et al., 2016; Shapiro et al., 2018). Surprisingly, however, only very few studies have investigated the potential benefits of combining axial induction control and yaw control. Park and Law (2015), based on simplified wake models and advanced optimization techniques, show that significant power gains can be obtained by the combining static yaw and induction control but they do not analyze the respective effects of yaw and induction; furthermore, their optimal solutions in the aligned case converge to an underinductive operation mode for yawed turbines. Munters and Meyers (2018a, b) show, by means of adjoint

methods with full-state information and an actuator disk turbine model where wake rotation is neglected, that high power gains result from the combination of dynamic yaw and axial induction controls with Munters and Meyers (2018b) highlighting the potential of quasi-static yaw control in the (dynamic) overinductive regime. From these previous studies, thus, it is not clear if significant power gains could be realized in the overinductive regime when both the yaw and the axial induction control are static, nor to what extent the neglected wake rotation effects are important.

The second objective of the present study is therefore to ascertain if significant power gains can be obtained, with a combination of static yaw control and static axial induction control, by operating yawed turbines at higher axial induction and including the effect of wake rotation in the turbine model. An affirmative answer would allow to isolate the mean wake redirection as the most relevant physical effect at play (instead of e.g. the dynamical adaptation to the incoming wind) and that it is robust with respect to the inclusion of wake rotation effects. Furthermore, if successful, static overinductive yaw control could be easily implemented by simply updating existing yaw-control protocols with a prescription on the suitable turbine rotor-collective blade-pitch angle (controlling the axial induction and the thrust coefficient) for each accessible yaw angle.

The potential of static overinductive wake redirection will be investigated by computing power gains that can be obtained in a wind-turbine array composed of two spanwise-periodic rows of wind-aligned turbines where the same control is applied to all upwind-row turbines while downwind-row turbines are left in default operation mode. This idealized configuration, which is an extension to the spanwise-periodic case of the two-turbine configuration considered by Fleming et al. (2015), is chosen in order to keep simple the physical interpretation of the results by isolating the effects of tilt or yaw angle and axial induction of the upwind turbines without entering the problem of the optimization of these parameters encountered in more realistic configurations with more rows. As such, this approach is a necessary first step needed to isolate the main trends at play before considering more realistic settings. Importantly, the relevance of these power gains will be tested without excessive assumptions by means of large-eddy simulations in the atmospheric boundary layer using a turbine model which includes the effects of wake-rotation, radial force distribution and a realistic turbine controller.

We anticipate that substantial enhancements (up to a factor of 3) of the power gains induced by wake redirection are found when operating the tilted or yawed turbines at higher axial induction.

The formulation of the problem at hand is introduced in §2. Results are reported in §3 and further discussed in §4. Additional details on used methods are provided in Appendix A and additional results about the effect of using a less realistic turbine model, where wake rotation effects are neglected, are reported in Appendix B.

## 2   Problem formulation

We address the case of two spanwise-periodic rows of wind turbines immersed in a neutral atmospheric boundary layer (ABL) at latitude $41^o$N. The flow is simulated by means of large-eddy simulations (LES) with SOWFA (the Simulator for On/Offshore Wind Farm Applications developed at NREL, see Churchfield et al., 2012) which solves the filtered Navier-Stokes equations including the Coriolis acceleration associated to Earth's rotation and the compressibility effects modeled by means of the

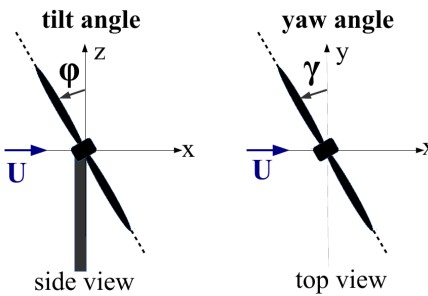

**Figure 1.** Definition of the positive rotor tilt and yaw angles $\varphi$ and $\gamma$ used in the present study. Positive tilt angles can be obtained for downwind-oriented rotors to avoid blade-tower hits.

Boussinesq approximation (see Appendix A for more details and Churchfield et al., 2012, for the explicit expression of the solved equations and a full description of the used formulation and modeling assumptions used in SOWFA).

NREL 5-MW turbines (Jonkman et al., 2009) are considered, which are modeled with SOWFA's native actuator disk method where wake rotation, the radial distribution of aerodynamic forces and the thrust coefficient are all computed from blade proper-
ties providing a reliable descriptions of the wake structure except in the near-wake region. We also make use of SOWFA's native implementation of NREL 5-MW's realistic five-region turbine controller based on generator-torque control in the Region-II regime corresponding to the mean wind speeds considered in the following; in this regime we modify axial induction by changing the rotor-collective blade-pitch angle $\beta$. Higher axial inductions are obtained by enforcing negative values of $\beta$ (see Appendix A), resulting in higher local thrust coefficients $C'_T = 2T/(\rho u_n^2 A)$, where $T$ is the thrust magnitude and $u_n$ is the
disk-averaged wind velocity component normal to the rotor disk of area $A = \pi D^2/4$. For all the considered cases the local power coefficient $C'_P = 2P/(\rho u_n^3 A)$ is well approximated as $C'_P = \chi C'_T$, with $\chi = 0.9$; results on $C'_P$ trends will, therefore, not be shown in the following. The incoming flow, generated by means of a precursor simulation in a 3km x 3km domain in the absence of turbines, has a 100m-thick capping-inversion layer centered at H=750m separating the neutral boundary layer with constant potential temperature ($\theta$=300K) from the geostrophic region above where the vertical potential temperature gradient
is positive $(d\theta/dz)_G = 0.03K/m$. In the capping-inversion layer this gradient is $(d\theta/dz)_{CI} = 0.03K/m$. In the precursor simulation, the ABL is driven by a pressure gradient adjusted to maintain a horizontally-averaged mean of 8m/s from the west at z=100m (a few meters above hub height $z_h$=89m). In the region spanned by the turbines ($z$ <152m) the streamwise mean velocity is well approximated by the logarithmic law and the vertical wind veer is less than $4^o$ (see Cossu, 2020b, where the same ABL has been already considered). The streamwise turbulence intensity of the incoming wind at hub height is of $5.7\%$
for the enforced low roughness length ($z_0 = 0.001$m) typical of offshore conditions.

Simulations in the presence of wind turbines are repeated in the same 3km x 3km domain starting from the solution of the precursor simulation at $t_0$=20000s, corresponding to a well developed ABL, up to $t_1$=30000s. Statistics are computed starting from $t$=24000s, when turbine wakes are fully developed. The pressure gradient issued from the precursor simulation is enforced during the simulation with turbines and the (previously stored) ABL solution at $x$=0 (west boundary) is used as
inflow boundary condition.

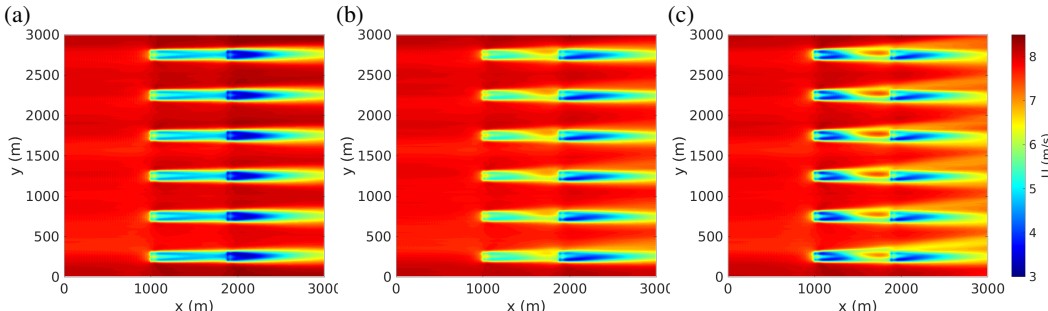

**Figure 2.** Tilt control: Mean (temporally averaged) streamwise velocity field in the horizontal plane at hub height obtained ($a$) in the baseline case where all turbines are operated in default mode, ($b$) with upwind turbines tilted by $\varphi = 30^o$ and operated at the default rotor-collective blade-pitch angle $\beta = 0^o$ and ($c$) with upwind turbines tilted by $\varphi = 30^o$ and operated at higher induction ($\beta = -5^o$). The mean wind is from the west (from the left, parallel to the $x$ axis). Note that the entire 3km x 3km computational domain is shown in the figure and that periodic boundary conditions are applied on the north and south boundaries.

In each (spanwise-periodic) row, turbines are spaced by $4D$ in the spanwise direction (where $D$=126m is the rotor diameter) and the two rows, are spaced by $7D$ in the streamwise direction with corresponding turbines of each row aligned with respect to the mean-wind direction (see Fig. 2, where the full computational domain is shown). Downwind-row turbines are always operated in default mode with the rotor axis at zero yaw angle $\gamma = 0^o$ (aligned with the mean wind at z=100m), tilt angle $\varphi = -5^o$ to prevent rotor-tower hits (see Fig. 1 for a definition of $\varphi$ and $\gamma$) and rotor-collective blade-pitch angle $\beta = 0^o$. In the baseline (reference) case upwind-row turbines are also operated in default mode. The baseline case is then compared to a set of controlled cases where all the turbines of the upwind row are operated at the same non-zero tilt or yaw angle and, possibly, non-zero rotor-collective blade-pitch angle.

## 3 Results

### 3.1 Effect of overinduction on tilt control

In the baseline case (all turbines operated with $\gamma = 0^o$, $\varphi = -5^o$, $\beta = 0^o$), the usual situation is found where the turbines of the downwind row see a strongly reduced mean wind (see Fig. 2$a$ and Fig. 3$b$) therefore producing only ≈30% of the total power, i.e. ≈40% of that produced by the upwind row of turbines (see Fig. 4$b$). The effect of wake rotation is clearly discernible in the mean streamwise vorticity field (see Fig. 3$a$). In the following, power gains will be computed with respect to the mean power $P_{Ref}$ produced in this baseline case.

We then consider the case where upwind-row turbines are tilted by $\varphi = 30^o$, an angle in the range where best power gains have been found in previous studies (Fleming et al., 2014, 2015; Cossu, 2020a, b), while keeping their rotor-collective blade-pitch angle at the default value $\beta = 0^o$. In this case, the wakes of the upwind turbines are pushed down by the tilt-induced downwash increasing the mean wind available to downwind turbines (see Fig. 2$b$). The tilt-induced decrease of power produced

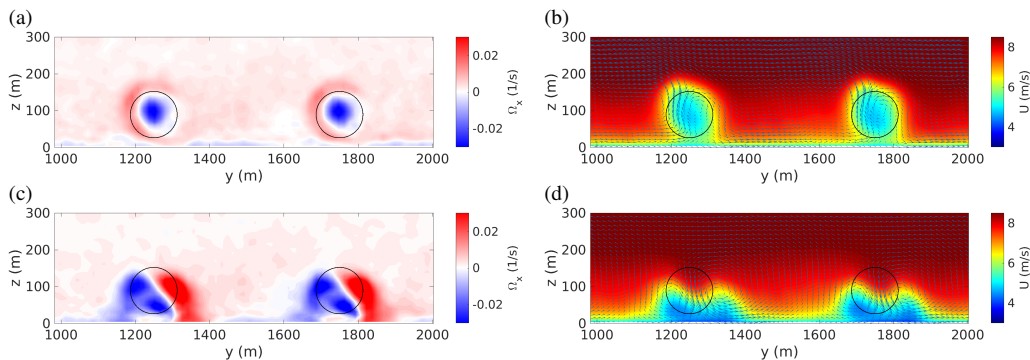

**Figure 3.** Tilt control: Cross-stream view of the mean streamwise vorticity and velocity fields in the in the baseline case (top panels $a$ and $b$) and with upwind turbines tilted by $\varphi = 30^o$ and operated at $\beta = -5^o$ (bottom panels $c$ and $d$). From the streamwise vorticity fields (left panels $a$ and $c$), extracted $3D$ downstream of the first turbine row, the negative streamwise vorticity in the wake core associated to wake rotation can be clearly seen in the baseline case (panel $a$) as well as its combination with the two counter-rotating streamwise vortices forced by the tilted rotor (panel $c$). Streamwise (color scale) and cross-stream (arrows) velocity fields (right panels $b$ and $d$) are extracted $D/2$ upstream of the second row of turbines; to improve readability only the fields of the two central turbines columns (between $y = 1000m$ and $2000m$) are shown. The circles in black represent the perimeter of downstreamn rotors.

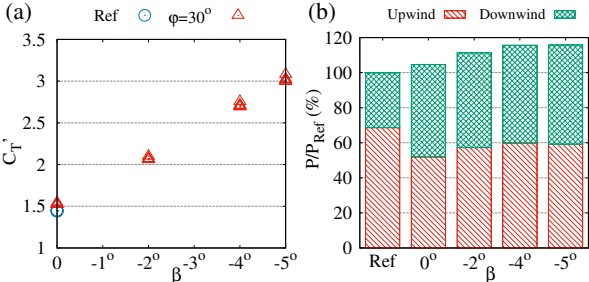

**Figure 4.** Effect of enforcing negative rotor-collective blade-pitch angles $\beta$ on upwind-row turbines tilted by $\varphi = 30^o$. Panel $(a)$: (temporally-averaged) local thrust coefficient $C'_T$ of the individual turbines of the upwind row. Panel $(b)$ wind power extracted by the upwind (hatched red) and downwind (cross-hatched green) rows of turbines normalized by the total power $P_{Ref}$ produced in the baseline case (Ref).

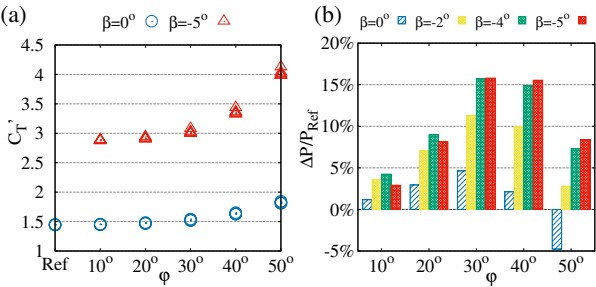

**Figure 5.** Effect of the tilt angle $\varphi$ on: (*a*) the local thrust coefficients $C_T'$ of upwind-row turbines when they are operated with $\beta = 0^o$ (default axial induction) or with $\beta = -5^o$ (strongly overinductive regime), (*b*) the total power gain $(P - P_{Ref})/P_{Ref}$ for selected values of $\beta$.

by upwind-row turbines is compensated by the increase of the power produced by downwind-row turbines resulting in global power gains of $\approx 5\%$ for $\varphi = 30^o$ (see Fig. 4*b*).

In a further step, the rotor-collective blade-pitch angle of the tilted upwind-row turbines is changed. Enforcing increasingly negative values of $\beta$ (i.e. increasing the mean angle of attack of all rotor blades, as explained in Appendix A) results in increased thrust coefficients (increased axial induction) which, starting from $C_T' = 1.5$ in the baseline case ($\beta = 0^o$), attain $C_T' = 3$ for

$\beta = -5^o$ in turbines tilted by $\varphi = 30^o$ (see Fig. 4*a*).

The effect of the increased thrust is twofold: (a) the downwash associated to the stronger tilt-induced streamwise vortices is reinforced (see Fig. 3*c,d*), which increases the mean wind speed seen by downstream rotors (see Fig. 2*c* and Fig. 3*d*) and their extracted power despite the higher wake deficit of upwind turbines (compare Fig. 2*c* to Fig. 2*b*) and (b) the power produced by tilted turbines is also (slighlty) increased[1] (see Fig. 4*b*). The combination of these two effects results in optimal power gains

which are highly enhanced (almost tripled) with respect to those obtained by tilt without overinduction.

Finally, a full set of $\varphi$-$\beta$ combinations is considered. For these simulations we observe that, for turbines operated at constant $\beta$, the increase of $C_T'$ with $\varphi$ is noticeable only for $\varphi \gtrsim 30^o$, as shown in Fig. 5*a* (we have verified that this increment is consistent with the effects of changing the tilt angle and the associated change of the induction factor). Considering the $(P - P_{Ref})/P_{Ref}$ power gains with respect to the baseline case, from Fig. 5*b* it can be seen that the maximum power gains are

reached for $\varphi \approx 30^o$ with optimal values obtained with significant overinduction (power gains larger than $15\%$ for $\beta \approx -5^o$) which are almost three times those ($\approx 5\%$) obtained with tilt control at reference induction rates ($\beta = 0^o$). This effect of overinduction in tilt control is very strong: from Fig. 5*b* it is indeed also seen that at $\varphi = 30^o$, even with the moderate rotor-collective blade-pitch angle $\beta = -2^o$ power gains have already almost doubled with respect to standard tilt control with $\beta = 0^o$.

The high enhancement of power gains obtained by combining overinduction with tilt control with respect to those obtained

by standard tilt control at baseline induction is consistent with that found in our previous studies (Cossu, 2020a, b) therefore

---

[1] This might be related to blockage effects which induce an increase with $C_T'$ of the power produced by an (upwind) spanwise-periodic row of turbines as shown by Strickland and Stevens (2020) and it is not surprising given that for the NREL5 turbine $\beta = 0^o$ corresponds, by design, to the maximum $C_P$ (at the optimal wind-tip speed ratio) for an isolated non-tilted turbine but not necessarily so when $\varphi = 30^o$.

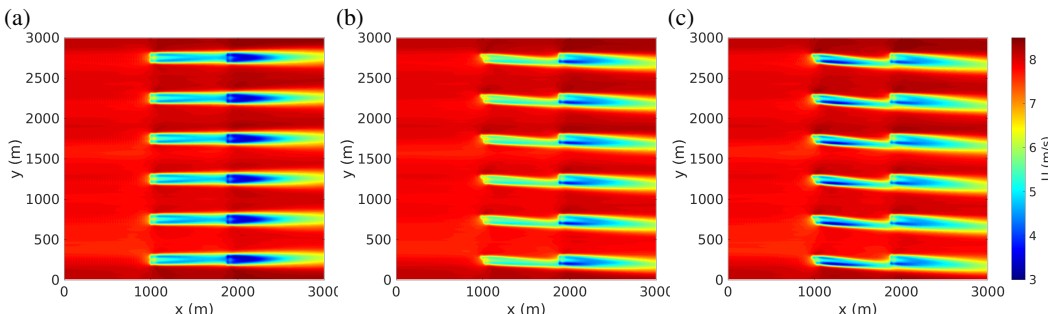

**Figure 6.** Yaw control: Mean streamwise velocity field in the horizontal plane at hub height obtained (*a*) in the baseline case where all turbines are operated in default mode ($\gamma = 0^o$, $\beta = 0^o$ same as Fig. 2*a*, reproduced here to ease the comparison), (*b*) in the case with upwind turbines yawed by $\gamma = 30^o$ and operated at the default $\beta = 0^o$ and (*c*) with upwind turbines yawed by $\gamma = 30^o$ and operated at higher induction ($\beta = -4^o$).

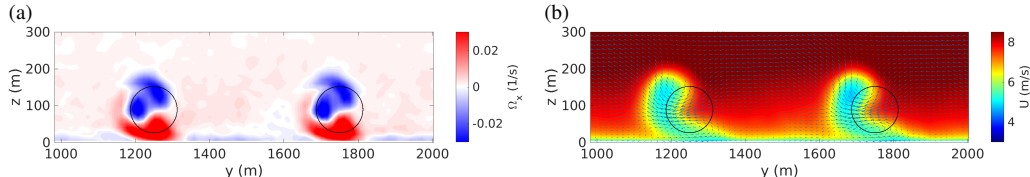

**Figure 7.** Yaw control: Cross-stream view of the mean streamwise vorticity and velocity fields with upwind turbines yawed by $\gamma = 30^o$ and operated at $\beta = -4^o$. The signature of the two vertically-staked counter-rotating streamwise vortices forced by the yawed rotor combined with wake rotation is clearly visible in the streamwise vorticity field (panel *a*) extracted $3D$ downstream of the first turbine row. Their effect on the lateral displacement of the wake is clearly discernible in the streamwise (color scale) and cross-stream (arrows) velocity fields (panel *b*) extracted $D/2$ upstream of the second row of turbines. Only the fields of the two central turbines columns (between $y = 1000m$ and $2000m$) are shown.

confirming the robustness of this trend. The absolute levels of power gains are, however, smaller than those reported by (Cossu, 2020a, b) both because two-rows arrays are considered here instead of the previously considered three-rows arrays (which have higher power gais, see e.g. Annoni et al., 2017) and because wake-rotation effects, neglected in the previous studies, are here taken into account (see Appendix B for further details).

## 3.2 Effect of overinduction on yaw control

We now evaluate the benefits of combining static yaw control with static overinduction. We proceed similarly to the tilt-control case by using the same precursor simulation and the same baseline case where all turbines operate at default values $\gamma = 0^o$, $\varphi = -5^o$, $\beta = 0^o$.

We first simulate the standard yaw control where the yaw angle $\gamma$ of upwind-row turbines is changed (while keeping unchanged the other parameters $\varphi = -5^o$, $\beta = 0^o$) resulting in the well known horizontal deviation of upwind-row turbine wakes and the increase of the mean wind speed seen by downwind rotors (see Fig. 6*b*). From Fig. 8*b* it is seen that the increase of

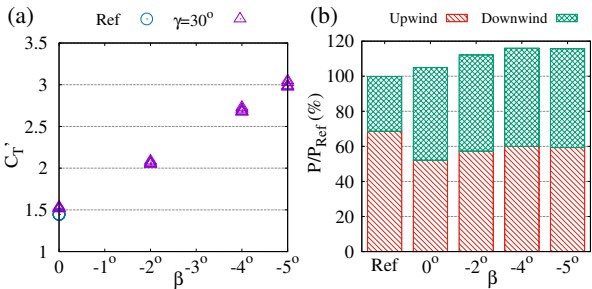

**Figure 8.** Effect of changing the rotor-collective blade-pitch angle $\beta$ of turbines yawed by $\gamma = 30^o$. Panel $(a)$: local thrust coefficient $C_T'$ of the turbines of the upwind row. Panel $(b)$: wind power extracted by the upwind (hatched red) and downwind (cross-hatched green) rows of turbines normalized by the total power $P_{Ref}$ extracted in the baseline case.

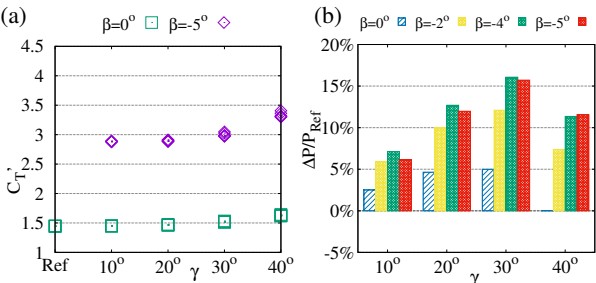

**Figure 9.** Effect of the yaw angle $\gamma$ on: $(a)$ the local thrust coefficients $C_T'$ of upwind-row turbines when they are operated at $\beta = 0^o$ or at $\beta = -5^o$, and $(b)$ power gains for selected values of rotor-collective blade-pitch angle $\beta$.

the power produced by downwind-row turbines compensates the reduction of the power produced by the yawed (upwind-row) resulting in maximum power gains of $\approx 5\%$ obtained for $\gamma \approx 30^o$, similarly to the values found by Fleming et al. (2015) for the two-turbines case.

Increasing the local thrust coefficient $C_T'$ by means of increasingly negative blade-pitch angles in yawed turbines (see Fig. 8$a$) has effects similar to those observed for the tilt-control case: an increase of velocity deficits in upwind-row turbine wakes but also their higher deviation away from downwind turbines (see Fig. 6$c$ and Fig. 7$b$) induced by the stronger yaw-induced vertically-staked counter-rotating streamwise vortices (see Fig. 7$a$) resulting in an increase of the mean power produced by all turbines with respect to the standard yaw-control case with $\beta = 0^o$ (Fig. 8$b$).

The analysis of a full range of $\gamma$-$\beta$ combinations, leads to results similar to those obtained for the tilt-control case. A non-negligible increase of $C_T'$ is observed for large yaw angles $\gamma \gtrsim 30^o$ when operating at constant $\beta$, as reported in Fig. 9$a$, and global power gains obtained by yaw control are highly enhanced when yawed turbines are operated at higher induction (more negative values of the rotor-collective blade-pitch angle $\beta$). Also similarly to the tilt-control case, maximum power gains are obtained for $\gamma \approx 30^o$ regardless of the $\beta$ value. Overall optimal power gains (above 15%) are reached for relatively high
overinduction ($\beta \approx -4^o$). Also in this case, power gains obtained by $\gamma = 30^o$ yaw control are more than doubled already for

$\beta = -2^o$ and almost tripled for the optimal value $\beta = -4^o$ with respect to the standard operation mode ($\beta = 0^o$) at the same yaw angle $\gamma = 30^o$.

These results confirm the first intuition that, also in the static yaw-control case, static overinduction leads to a substantial improvement of the power gains which is based on the same mechanisms discussed for the tilt-control case confirming that these mechanisms are quite robust.

## 4  Conclusions

The main goal of this study was to assess the magnitude of global power gains that can be obtained in wind turbine arrays by combining static wake redirection control and static axial induction control operating tilted or yawed turbines at higher axial induction (overinduction). Results have been obtained by means of large-eddy simulations of a two-rows array of NREL 5MW turbines in a neutral atmospheric boundary layer.

In a first part of the study we have considered the effect of higher induction on tilt-control by using an actuator disk model less idealized than the one used in our previous studies of this approach. The results confirm that, also with this more realistic turbine model, power gains can be highly increased by operating tilted turbines at higher induction (power gains above 15% are found, to be compared to ≈5% obtained with default induction, for the considered set of parameters). This substantial enhancement of power gains due to the use of overinduction in tilt control is consistent with those found in our previous studies but the absolute levels of the power gains are smaller because of the differences in array configurations and in the used turbine models. Indeed, when included in the turbine model, wake rotation results in an inclination of the formerly vertical downwash which displaces higher-altitude higher-speed fluid towards downstream rotors and, as a consequence, a decrease of tilt-induced power gains.

In the second part of the study we have ascertained if similar power gain enhancements could be obtained by combining static overinduction with static yaw control. To this end, we have first considered the standard case where yawed turbines are operated at the reference rotor-collective blade-pitch angle $\beta = 0$ finding power gains of the order of 5%, similar to those found in many previous studies (e.g. Fleming et al., 2015, for the two-turbines case). We then show that a very significant increase of power gains (almost threefold, up to ≈15% for the cases considered) is obtained by operating yawed turbines at higher induction, similarly to what found for tilt control.

The findings concerning the static overinductive yaw control are probably the most relevant of this study for short-term applications because they show that significant power gains can be realized with a simple static overinductive yaw control in a realistic model (the atmospheric boundary layer with NREL 5-MW turbines simulated with SOWFA) where wake rotation effects are fully taken into account. They also probably isolate the main physical mechanisms underlying the significant power gains found by Munters and Meyers (2018b, a) by means of combined (dynamic and static) yaw and (dynamic) induction control using adjoint methods with full state information on large-eddy simulations where the turbines were modeled with a simplified actuator disk method neglecting wake rotation effects. Furthermore, static overinductive yaw control is suitable for

immediate experimental testing with most existing standard horizontal axis wind turbines unlike tilt control which is promising for specifically designed future generation downwind-oriented and/or floating turbines (Bay et al., 2019; Nanos et al., 2020).

Another important result, obtained for both tilt and yaw overinductive controls, is that while maximum power gains ($\approx 15\%$) are obtained for relatively large rotor-collective blade-pitch angle ($\beta = -5^o$) for the optimal large tilt and yaw angles ($\varphi, \gamma \approx 30^o$), significant power gains ($\approx 10\%$) are already obtained for smaller values $\beta = -2^o$ showing the robust beneficial effect effect of even moderately overinductive turbine operation.

It is also to be noted that here we have considered only two rows of turbines and for a single configuration with a small value
of the $D/\delta$ ratio of rotor diameters to the ABL thickness but that higher power gains can be expected for a larger number of turbine rows (Park and Law, 2015; Annoni et al., 2017; Cossu, 2020a) and for larger values of $D/\delta$ (Cossu, 2020a, b).

Additional investigations are, however, necessary to further refine, in many directions, the conclusions of the present study. A first important issue is to understand what are the effects of overinduction on the static and dynamic structural loads experienced by the blades of tilted and yawed turbines. A complete aeroelastic analysis based on higher-fidelity simulations making use of
the actuator line method, requiring more refined grids and time steps and larger computational resources, is highly desirable, especially for the largest considered values of the yaw, tilt and pitch angles where the near- and middle-wake structures are probably more sensitive to details of the turbine model.

Other issues are wind direction and array configuration. The present study is limited to a two-rows array in the wind-aligned case, but it is, of course, important to evaluate power gains in arrays with many more rows also in non-aligned configurations.
Such kind of analysis, where the optimal combination of tilt, yaw and pitch angles of all turbines has to be computed for a high number of wind directions and intensities, would be too computationally demanding if performed by means of large eddy simulations and is customarily based on less computationally demanding simplified sets of equations where the accurate modeling of the controlled wakes is of primary importance (see e.g. Boersma et al., 2017). In this context, the results presented in the present study could be used to help in the improvement and validation of simplified wake models in moderate to high
tilt/yaw- and pitch-angle regimes, particularly in the case of significant overinduction. Such improved models would allow for more reliable predictions of annual energy production gains obtained with overinductive yaw or tilt control for realistic wind roses and wind farm configurations by using advanced optimization methods such as those used by Park and Law (2015).

Finally, it would be very interesting to ascertain if additional power gain enhancements could come from the simultaneous activation of tilt, yaw and axial induction control. It might indeed be possible that, as a consequence of the symmetry breaking
associated to wake rotation effects and Coriolis acceleration, optimal power gains are obtained with "hybrid" yaw-tilt rotor-axis rotations even in wind-aligned configurations. This is the subject of current intense research effort.

## Appendix A: Methods

The large-eddy simulations presented in this study are performed with SOWFA, a set of libraries and codes able to simulate atmospheric flows over wind turbines (Churchfield et al., 2012), that is based on the OpenFOAM software environment de-
signed to solve partial differential equations by means of finite-volume spatial discretizations on unstructured meshes (Jasak,

2009; OpenCFD, 2011). The filtered Navier-Stokes equations are solved using the Smagorinsky (1963) model to approximate subgrid-scale stresses with compressibility effects accounted for by means of the Boussinesq approximation and Earth's rotation effects accounted for by the Coriolis acceleration term in the equations (see Churchfield et al., 2012, for all details on the used formulation and for a validation of the code in the atmospheric boundary layer). Schumann (1975) stress boundary conditions, modeling the effect of ground roughness, are applied near the ground and slip boundary conditions are enforced at at the top of the solution domain. The solutions are advanced in time using the PIMPLE scheme.

Periodic boundary conditions are applied in the $x$ (west-east) direction for the preliminary 'precursor' simulations where the atmospheric boundary layer flow is computed in the absence of wind turbines in order to generate realistic inflow wind conditions (Keating et al., 2004; Tabor and Baba-Ahmadi, 2010; Churchfield et al., 2012). The mean pressure gradient is adapted in order to maintain (horizontally-averaged) mean westerly winds of 8m/s at $z = 100m$. The time-history of the mean pressure gradient and of the solution at $x = 0$ are stored and then used in the simulations with wind turbines which are run in the same domain with the same grid but removing the periodicity constraint in the streamwise direction and replacing it with an inflow condition enforcing the solution found $x = 0$ in the precursor simulation. Periodic boundary conditions are applied in the $y$ (south-north) direction for both precursor simulations and simulations with turbines.

The solution domain extends 1km in the vertical direction and 3km x 3km along the $x$ and $y$ axes and is discretized with cells extending 15m x 15m in the $x$ and $y$ directions and $7m$ (near the ground) to $21m$ (near the top boundary) in the vertical direction. $\Delta t = 0.8s$ time steps are used to advance the solution. These parameters keep manageable the amount of data stored in the precursor simulation.

The aerodynamic forces developing on NREL 5-MW turbines, having a $D$=126m rotor diameter and $z_h$=89m hub height (Jonkman et al., 2009), are modeled with SOWFA's native actuator disk method based on the blade-element method (BEM). The forces exerted on the fluid are computed for each radial blade section by using the lift and drag coefficients $c_L(\alpha)$, $c_D(\alpha)$ associated to the local NREL 5-MW blade profiles and the local angle of attack $\alpha = \phi - (\theta + \beta)$ computed as the difference between the angle $\phi$ formed by the relative wind seen by the blades with the rotor plane and the local pitch angle which is the sum of the local twist angle $\theta$ of the blades and the rotor-collective blade-pitch angle $\beta$ (the reader is referred to e.g. Burton et al., 2001; Sørensen, 2011, for a detailed discussion of turbines modeling in general and of the BEM in particular). The Gaussian projection of the discretized body forces proposed by Sørensen and Shen (2002) is also used with a smoothing parameter $\varepsilon = 20m$ to avoid numerical instabilities (Martínez-Tossas and Leonardi, 2013).

The NREL 5-MW five-region controller implemented in SOWFA is used to control the turbines rotational speed and axial induction. In the Region II regime, the one accessed in the presented simulation, the turbine is driven to the design point (tip-speed ratio and thrust coefficient corresponding to the maximum power coefficient for an isolated non-tilted non-yawed turbine) by means of generator-torque control at the default rotor-collective blade-pitch angle $\beta = 0^o$. In this regime, we enforce the axial induction control by changing the rotor-collective blade-pitch angle $\beta$ while leaving unchanged the other parameters of the generator torque controller.

The local thrust coefficient is retrieved from the computed turbine thrust magnitude and rotor-averaged normal mean wind speed $u_n$ by making use of its definition $C'_T = 8T/\pi\rho u_n^2 D^2$.

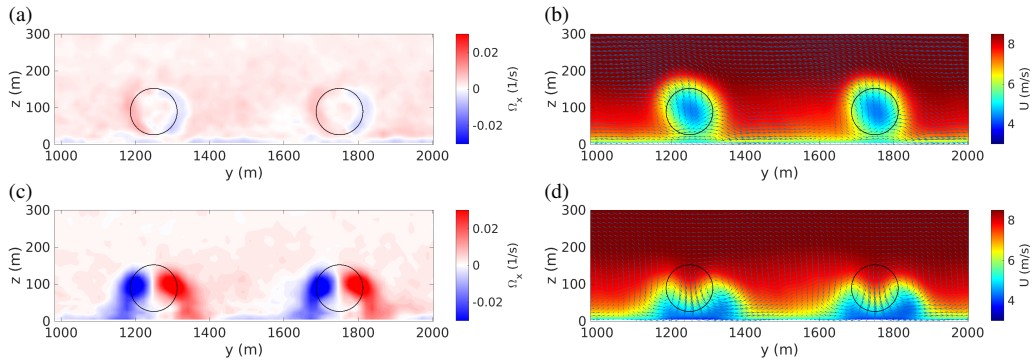

**Figure B1.** Tilt control: Cross-stream view of the mean streamwise vorticity and velocity fields obtained by using the ADMC turbine model in the baseline case where all turbines are operated at $C'_T = 1.5$ with no tilt or yaw (top panels $a$ and $b$) and with upwind turbines tilted by $\varphi = 30^o$ and operated at $C'_T = 3$ (bottom panels $c$ and $d$). The streamwise vorticity fields (panels $a$ and $c$) are extracted $3D$ downstream of the first turbine row, while the streamwise velocity fields (panels $b$ and $d$) are extracted $D/2$ upstream of the second row of turbines.

## Appendix B: Effect of the used turbine model on tilt-control

A quantitative analysis of the effect of the improved ADM model used in the present study by means of a direct comparison with the results obtained in Cossu (2020b) is not possible due to the difference of the considered array configurations (two arrays here, three in Cossu, 2020b). Additional simulations of tilt-control have therefore been performed by using the same turbine model (ADMC) used in Cossu (2020b) for the same array configuration used in the present study. We recall that, contrary to SOWFA's ADM used in the present study, in the ADMC model wake rotation effects are neglected and a uniform load is assumed over the rotor disk that is assumed to operate at constant $C'_T$.

First a baseline case has been simulated with all turbines operated at the reference values $C'_T = 1.5$, $\varphi = -5^o$, $\gamma = 0^o$. Then, a standard tilt-control case has been considered with upwind-row turbines operated at $C'_T = 1.5$, $\varphi = 30^o$ (and $\gamma = 0^o$) obtaining a power gain $\Delta P / P_{Ref} \approx 11\%$. Finally, overinductive tilt control has been tested by operating at $C'_T = 3$ the upwind row turbines tilted by $\varphi = 30^o$ obtaining a power gain of $\approx 27\%$.

For the 2-rows array layout, therefore, the ADMC model also predicts that power gains obtained by overinductive tilt control are much larger than those obtained by standard tilt control (by a factor of $\approx 240\%$ for the ADMC turbine model and by a factor of $\approx 330\%$ with SOWFA's ADM for $\varphi = 30^o$). However, the absolute levels of power gains computed with the ADMC model are higher than those computed with SOWFA's ADM turbine model. In this context, the effect of wake rotation appears to be important. In the ADMC model which applies a uniformly distributed force purely normal to the rotor disk, wake rotation effects are indeed neglected, resulting in a negligible mean axial vorticity in the rotor wake in the baseline case and in almost-symmetric counter-rotating vortices in the tilted case (see Fig. B1$a$ and $c$). In the ADMC tilted case, therefore, the downwash associated to the tilt-induced streamwise vortices is purely vertical resulting in a highly efficient displacement of higher-altitude higher-momentum fluid towards the downstream-rotor swept area (see Fig. B1$d$). In the case of the SOWFA's ADM more realistic turbine model, on the contrary, wake rotation effects are fully taken into account, resulting in non-negligible

mean axial vorticity in the rotor wake in the baseline case and in strongly non-symmetric counter-rotating vortices in the tilted case (see Fig. 3$a$ and $c$). In the more realistic case, therefore, the tilt-induced streamwise vortices are associated to an oblique downwash which is less efficient in displacing high-momentum fluid towards the downstream rotors (see Fig. 3$d$). This explains that lower absolute values of tilt-induced power gains are obtained when wake-rotation effects are taken into due account.

*Competing interests.* The author declares no conflict of interest.

*Acknowledgements.* I gratefully acknowledge the use of the Simulator for On/Offshore Wind Farm Applications (SOWFA) developed at NREL (Churchfield et al., 2012) based on the OpenFOAM finite volume framework (Jasak, 2009; OpenCFD, 2011).

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
