# Peer review of "Wake redirection at higher axial induction"

_Wind Energy Science, 2020_

## Referee Comment (RC1) · Wim Munters (Referee) · 22 Nov 2020

In the manuscript "Wake redirection at higher induction", the author describes a study into combining wake redirection techniques from yaw and tilt control with increased turbine induction. The study is well-described and the structure and elaboration of the manuscript is very clear and easy to read. The overall contribution to the field is rather limited and incremental, i.e. tilt control at higher induction has been shown in an earlier study of the same author (albeit using a different turbine model); and combined yaw and induction control has been shown in earlier studies by Park Law and Munters Meyers (albeit using different ways of generating control strategies). That being said, the current work is still highly relevant to the general community and I believe the topic is suitable for publication in Wind Energy Science. However, I feel there are several points for imprfovement of the quality and novelty of the considered work, as detailed below in my comments.

[Figure]

**Major comments**

- The added contribution of the current paper is relevant but incremental: overinduction has already been shown to work for yaw and tilt control in earlier LES-based studies (Cossu 2020b and Munters and Meyers 2018 respectively. The author shows that this strategy also works in his current setup (with a slightly different turbine model for tilt, and a static vs dynamic control strategy for yaw). The added value of the current paper over existing literature would benefit from a more detailed flow analysis of the current LES results. For example, it would be interesting to see expand Figure 1 with additional flow field sections and compare to results from Cossu 2020b, which would allow to show effects of wake rotation on tilt-based redirection. Further, a flow-based comparison between the differences for yaw and tilt control would be very interesting. For example, the author mentions in line 160 that the shift of larger optimal angles after including pitch is present for tilt but not for yaw, and that this can 'probably' be explained by observing that vertical shear is not exploited by yaw. The author has the data to show this quantitatively, and I feel this could be an important addition to the current work.

- The first goal of the study is to assess whether additional gains in overinductive tilt control still hold up when considering realistic turbine models closer to reality. However, I believe that this goal is only partially achieved and the step forwards from the Cossu 2020b study is relatively small. A significant step forward would have been made using an actuator line model instead of an actuator disk model. The limitations of an actuator disk model should be mentioned earlier in the study (currently they are left to the conclusions). Some comments related to this:

  - In the conclusion, the author mentions that 'the absolute level of power gains is larger in Cossu 2020a, b'. Unless I'm mistaken, this is not mentioned in the main text. The author should attempt to explain this. Could this be due to

the different turbine model (e.g. accounting for wake rotation), or a different wind-farm setup (i.e. 2 rows vs 3 rows)?

- The increase of $C'_T$ shown in Figure 3b requires some further explanation, is this caused by a change in the effective angle of attack of the blades?
- The author shows the dependency of $C'_T$ on different control parameters, but there is no mention of how e.g. the pitch angle affects the power coefficient $C_P$ (or $C'_P$ if you will). This should be clearly mentioned

• The author frequently mentions achieving 'doubled' or 'tripled' power gains in high induction compared to baseline tilt/yaw control. Please be more specific in phrasing here to avoid confusion: mention explicitly the percentages, and the setup (e.g. Cossu 2020b has a three-row setup, achievable power gains are different than when looking at two rows as in the current study).

• Starting from line 131, the author discusses that he believes increasing thrust in tilted conditions should not impact turbine loading compared to standard operation, since the overall thrust force would not be higher than in the latter. However, Fleming et al (Renewable Energy 2014), have shown that tilt control can have a significant influence on blade bending and drivetrain torsion. Further increasing thrust could aggravate such issues. I believe that turbine loading could be an issue at higher induction scenarios such as considered here, and that conclusive statements warrant a detailed analysis using aero-elastic codes. This should be mentioned in the manuscript.

**Minor comments**

• Line 59: typo oveinductive should be overinductive

• Line 82, the formula for $C'_T$ contains a $\pi$ which shouldn't be there

- The appendix states use of a Schumann BC at the wall. What is the roughness length imposed at the bottom and, more importantly, what is the resulting turbulence intensity at turbine height? This tends to have a significant impact on power deficits and hence achievable gains.

- A 3x3 km periodic precursor domain is probably too small to generate fully realistic turbulent flow structures. Does the author expect this to affect results in any way?

- The author rightfully mentions surprisingly little research efforts into combining yaw and induction control. An additional study that could be mentioned here is "Munters, Meyers, 2018, Optimal dynamic induction and yaw control of wind farms: effects of turbine spacing and layout. J Phys Conf. Ser 1037, 032015", which investigates combined dynamic yaw and overinduction for a series of different wind-farm layouts

---

## Referee Comment (RC2) · Anonymous Referee #2 · 24 Nov 2020

**Review of "Wake redirection at higher axial induction"**
**Author: Carlo Cossu**
The manuscript performs series of large-eddy simulations of a two-row wind farm in order to investigate the effect of yaw and tilt controls on overall performance of the wind farm. The manuscript is well written. However, it lacks several necessary details about the simulations in Section 2. The manuscript also does not provide any significant discussion about impact of control on the flow physics. But, the main concern of the reviewer is that it has only considered two-row wind farm, and it is not clear how this work can be linked to larger wind farms with more rows.
    The author is asked to address the following comments in the revised submission.

**Specific comments:**

1. The abstract mainly provides general introduction and the past research of the author. Please discuss the findings of the current research. You can quantify and further discuss the power gains due to tilted rotor and yaw control.

2. It is not clear why you have considered just two rows of turbines. I do not see any technical challenge in simulating for wind farms with more rows. But in case you think two-row wind farm simulation is sufficient, please discuss how you can link your findings to larger wind farms.

3. Line 81 to 83: Please add more explanation to clarify the relation between $\beta$ and $C_T'$. You can use blade element momentum (BEM) theory in order to describe the relations between $\beta$, lift and drag coefficients and the thrust coefficient.

4. Line 84 to 91: Add LES and other relevant equations. Adding a schematic for the computation domain will be helpful too.

5. This may be beyond the scope of this manuscript, but how practical do you think it is to tilt blade by $\phi = -30°$? Higher tilt angle will significantly increase the flapwise bending and reduce the blade lifetime. You have mentioned about gravity load in line 137, but that is not very clear.

6. Line 116 to 122 and Figure 3 (a): I do not understand why increasing $\beta$ (making it more negative) increases the power from the first turbine row. Wind turbines are usually optimized for the pitch angle around $0°$. If that is the case with your turbine too, power output should be lower for $\beta < 0°$. Increased thrust coefficients–for negative blade pitch angles–are simply caused by increased drag coefficients, and they will not necessarily translate into the higher power output.

7. You have not discussed how the tilt control and the yaw control influence the flow fields inside the wind farm. How do turbulence fields and shear stresses change as a result of those controls should be presented.

**Minor comments and corrections:**

1. Line 10: an high → a high.

2. Line 8: of the produced → of that produced

3. Line 110: Is it $\phi = -30°$? You can add a schematic describing positive and negative directions for yaw, tilt and pitch angles.

---

## Author Comment (AC1) · 13 Jan 2021

Please find the reply and the highlighted revised manuscript in the supplement zip file.

Please also note the supplement to this comment:
https://wes.copernicus.org/preprints/wes-2020-111/wes-2020-111-AC1-supplement.zip

---

## Author Comment (AC2) · 13 Jan 2021

Please find the reply and the highlighted revised manuscript in the supplement zip file.

Please also note the supplement to this comment:
https://wes.copernicus.org/preprints/wes-2020-111/wes-2020-111-AC2-supplement.zip

———————————————

---

## Author Response (AR1)

Carlo Cossu
*Directeur de recherche CNRS*
*Head of the Urban & Coastal Atmospheric Dynamics Group*
LHEEA, CNRS - Centrale Nantes
1 rue de la Noë, 44300 Nantes, France
☎ +33(0)240376824, ✉ carlo.cossu@ec-nantes.fr

[Figure]

[Figure]

Prof. Johan Meyers
Associate Editor
*Wind Energy Science*

Nantes, 13 January 2021

Dear Pr. Meyers,

I thank you and the reviewers for having examined the manuscript entitled "Wake redirection at higher induction" and for the rapidity of the review process.

Both reviewers have raised a number of issues which have been addressed in the revised manuscript and in the reply to each reviewer. As you can read in the posted author's comments, I have followed most of the suggestions of Prof Munters (Reviewer 1) while I found it more difficult to follow all the suggestions of Reviewer 2. All the comments, nevertheless, have helped to improve the quality of the manuscript that I'm resubmitting for publication in *Wind Energy Science*.

During the revision process I have also realized that most of the turbine simulations were performed with an erroneous roughness length which was larger than the one used in the precursor simulation. I have rerun all the simulations with the correct value and updated the paper figures accordingly, but, luckily, the results do not change much.

Following the many reviewers' comments and suggestions, and the updated results from the new simulations, the manuscript has undergone a major revision, the main modifications being the following ones:

- All the presented results have been updated with the new simulations (with the correct $z_0$). The new results, reported in the revised manuscript, are mostly similar to the previous ones so that the main conclusions of the manuscript do not change. Changes resulting from these new simulations are updated in the revised manuscript.
- Additional simulation have been performed to analyze the role of the used turbine model, and in particular the effect of including wake rotation effects. The new results are presented and discussed in the new Appendix B and mentioned in the manuscript when appropriate.
- New figures have been added showing the mean streamwise vorticity and velocity fields in the cross-stream planes to highlight the role of the counter-rotating streamwise vortices forced by the tilt or yaw misalignement and discuss the role of wake rotation. A scheme has been added to define the tilt and yaw angles.
- The abstract and conclusions have been modified to make them more clear following reviewers' comments and suggestions.

All the modifications of the manuscript can be tracked in the highlighted  revised version of the manuscript which has also been posted (red = removed, blue = added or modified).

I hope that you and the reviewers will find this revised version suitable for publication.

Yours sincerely,

**Comments on the review of "Wake redirection at higher axial induction" - Reviewer 1, Wim Munters**

Carlo Cossu

Laboratoire d'Hydrodynamique Énergetique et Environnement Atmosphèrique (LHEEA)
CNRS - Centrale Nantes, 1 rue de la Noë 44300 Nantes, France

January 13, 2021

I thank Prof. Munters for the many constructive comments and suggestions and I appreciate the rapidity of the refereeing process. Each issue raised by a specific comment is addressed in detail below.

This reply has taken longer than expected because during the revision, in relation to the performed additional simulations and to the question on the used roughness length, I became aware that most turbine simulations input files were bugged leading to the inconsistent use of roughness lengths $z_0 = 0.15$ that were larger than the value used in the precursor simulation. I therefore had to rerun all the simulations with the correct value $z_0 = 0.001$, postprocess the results and redraw all the figures.

The manuscript has undergone a major revision, the main modifications being the following ones:

- All the presented results have been updated with the new simulations (with the correct $z_0$). The new results, reported in the revised manuscript, are mostly similar to the previous ones so that the main conclusions of the manuscript do not change. Changes resulting from these new simulations are updated in the revised manuscript and are discussed, when appropriate, below.

- Additional simulation have been performed to analyze the role of the used turbine model, and in particular the effect of including wake rotation effects. The new results are presented and discussed in the new Appendix B and mentioned in the manuscript when appropriate.

- New figures have been added showing the mean streamwise vorticity and velocity fields in the crossstream planes to highlight the role of the counter-rotating streamwise vortices forced by the tilt or yaw misalignement and discuss the role of wake rotation. A scheme has been added to define the tilt and yaw angles $\varphi$ and $\gamma$.

- The abstract and conclusions have been modified to make them more clear following reviewers' suggestions.

———

*In the manuscript Wake redirection at higher induction, the author describes a study into combining wake redirection techniques from yaw and tilt control with increased turbine induction. The study is well-described and the structure and elaboration of the manuscript is very clear and easy to read. The overall contribution to the field is rather limited and incremental, i.e. tilt control at higher induction has been shown in an earlier study of the same author (albeit using a different turbine model); and combined yaw and induction control has been shown in earlier studies by Park Law and Munters Meyers (albeit using different ways of generating control strategies). That being said, the current work is still highly relevant to the general community and I believe the topic is suitable for publication in Wind Energy Science. However, I feel there are several points for improvement of the quality and novelty of the considered work, as detailed below in my comments*

I am glad that the study is found relevant and the topic suitable for publication in Wind Energy Science. While I agree with most of the comments, which have led to an improvement of the manuscript, I do not completely agree with the perceived lack of novelty/relevance of some of the results, and in particular those pertaining to the overinductive yaw control, as discussed below.

*The added contribution of the current paper is relevant but incremental: overinduction has already been shown to work for yaw and tilt control in earlier LES-based studies (Cossu 2020b and Munters and Meyers 2018 respectively. The author shows that this strategy also works in his current setup (with a slightly different turbine model for tilt, and a static vs dynamic control strategy for yaw).*

I agree that the contribution could be perceived as incremental in what concerns the tilt case. For the yaw-control results, however, I do not completely agree and I still am quite excited about the results. What is substantially new, relevant and the main step forward is, in my view, that the proposed overinductive yaw-control is static and, as such, immediately implementable in existing yaw-control settings. That this is the case, could not be clearly deduced from previous results with dynamic induction control where one could not exclude that the coordinated dynamical evolution of $C'_T(t)$ played a major role in the power gains. Also, the results of Park et al. (2015), which were obtained in the static-static case, seem to converge towards an underinductive regime for the front-row turbines in the wind-aligned case, probably because of their simplified wake model. Besides, the finding that a simple static open-loop overinduction control is able to significantly and systematically increase yaw-control power gains, and does so even for relatively small overinduction, also shows that the physics behind it is quite robust.

These points, not emphasized enough in the original manuscript, are more explicitly highlighted in the revised manuscript (lines 55-63 and 68-70).

*The added value of the current paper over existing literature would benefit from a more detailed flow analysis of the current LES results. For example, it would be interesting to see expand Figure 1 with additional flow field sections...*

I agree with this comment which has led to an improvement of the quality of the manuscript.

The flow analysis of the current results has been expanded by adding, in the additional figures 2,6 and B1 of the revised manuscript, cross-stream sections of the mean streamwise vorticity and velocity. These additional figures, and the associated discussion, show the effect of the tilt- and yaw-induced mean streamwise vortices and are allow to explain the important role of wake rotation in the newly added Appendix B.

*...and compare to results from Cossu 2020b, which would allow to show effects of wake rotation on tilt-based redirection.*

Thank you for this suggestion. As it is difficult to directly compare the present results to those of Cossu (2020b) because they are obtained for different array configurations (2 vs 3 rows), I have performed additional simulations with the same turbine model used in Cossu 2020b for the 2-rows layout considered in the present study. These results are discussed in the revised manuscript in the newly added Appendix B and mentioned in the main text (lines 154-159). It is, in particular, shown and discussed how the wake rotation induces a strong asymmetry of the induced streamwise vortices resulting in an oblique downwash and therefore in a loss of efficiency in the displacement of the high-speed fluid towards the downstream rotors.

*Further, a flow-based comparison between the differences for yaw and tilt control would be very interesting. For example, the author mentions in line 160 that the shift of larger optimal angles after including pitch is present for tilt but not for yaw, and that this can probably be explained by observing that vertical shear is not exploited by yaw. The author has the data to show this quantitatively, and I feel this could be an important addition to the current work.*

In the updated numerical simulations (using a consistent $z_0$) results, the mentioned shift to larger tilt angles is less pronounced than in the original manuscript. Furthermore, the newly discussed effect of wake rotation on the direction of the downwash in the tilt-induced case adds complexity to the interpretation because the shear direction to be considered is not vertical for the tilt case, which also probably introduces a further dependence on the streamwise turbine spacing. In the revised manuscript I have therefore removed the emphasis that was put on the shift to larger angles, as these shifts appear to be a relatively minor and probably non-generic effect.

*The first goal of the study is to assess whether additional gains in overinductive tilt control still hold up when considering realistic turbine models closer to reality. However, I believe that this goal is only partially achieved and the step forwards from the Cossu 2020b study is relatively small. A significant step forward would have been made using an actuator line model instead of an actuator disk model. The limitations of an actuator disk model should be mentioned earlier in the study (currently they are left to the conclusions).*

I do not completely agree on this point. This paper is mainly concerned with power gains in wind turbine arrays where turbines are not closely spaced. As these are time-averaged effects and they concern the interaction of far wakes with downstream rotors, I think that the actuator disk model is completely adapted to the purpose of this study and I do not think that any qualitative improvement would come from the use of at actuator line model. Furthermore, the high computational resources required in ALM simulations, which require grid refinements and much smaller time steps, would have restricted the analysis to a much more limited set of $\varphi - \gamma, \beta$ combinations.

The use of an ALM will of course be necessary for further highly suitable investigations of the structural loads and aeroelastic response of tilted and yawed turbines operated at higher induction. This is now mentioned in the revised manuscript (lines 225-227).

*Some comments related to this:*
*− In the conclusion, the author mentions that the absolute level of power gains is larger in Cossu 2020a, b. Unless Im mistaken, this is not mentioned in the main text. The author should attempt to explain this. Could this be due to the different turbine model (e.g. accounting for wake rotation), or a different wind-farm setup (i.e. 2 rows vs 3 rows)?*

This is an interesting suggestion which is closely related to an already discussed point.

In previous investigations of tilt-control it was found that power gains obtained in 3-turbines (3-rows) layouts were larger than those found in 2-turbines (2-rows) layouts, so this must certainly play a role in the difference in power gains. However, the respective role of the different number of rows and of the turbine model can not be isolated by a quantitative comparison of the present results with the mentioned previous ones, as both the array configuration and the turbine model have been changed. To isolate the role of the turbine model, additional simulations have thus been performed on the 2-rows configuration used in the present study but using the same turbine model (ADMC) used in Cossu 2020a,b (and in numerous previous investigations) as already mentioned in one of the previous points. These additional simulations show that the inclusion of wake rotation leads to a non-negligible decrease of the absolute level of power gains explaining the lower absolute values of tilt-induced power gains observed in the present study.

These additional results are reported in the newly added appendix B and are mentioned in the main text (lines 154-159).

*− The increase of $C'_T$ shown in Figure 3b requires some further explanation, is this caused by a change in the effective angle of attack of the blades?*

When the tilt angle $\varphi$ is changed at constant rotor collective blade pitch angle $\beta$ (and blade-twist $\theta$) the angle of attack $\alpha = \phi - (\beta + \theta)$ of the blades can change only because of a change in the angle $\phi$ formed by the wind with the rotor plane. The change in $\phi = \tan^{-1}(U_n/\Omega r)$, where $U_n$ is the velocity component normal to the rotor, is therefore a consequence of the change of $U_n$ and of the rotor angular speed $\Omega$ which not only depend on $\varphi$ but also on the load and torque on the turbine rotor (and therefore on $C'_T$) therefore forming a sort of closed loop. I therefore prefer to avoid explaining the change of $C'_T$ as caused by a change of $\alpha$. Furthermore, things are complicated by the fact that $\Omega$ is determined by the particular controller used for the specific turbine under consideration, that $U_n$ is not uniform on the disk and that wake rotation effects should also be included in the computation of $\phi$.

A probably simpler explanation (and also a check that the observed increase of $C'_T$ with $\varphi$ is not some kind of artifact) can be obtained by considering the direct dependence of $C'_T$ on $\varphi$ and on the change of the induction factor $a$ induced by the tilt. $C'_T$ can be expressed as a function of $\varphi$ and $a$ by e.g. rearranging Eq. (2.13) of Shapiro et al. (2018):

$$C'_T = \frac{4a}{(1-a)\cos^2 \varphi} \tag{1}$$

I have then plotted this predicted $C'_T$ in Fig. R1.1 as a function of $\phi$ by using axial induction factors $a$ based on the simulation data for $u_d$ as $a(\varphi) = u_d(\varphi)/(U_\infty \cos \varphi)$.

In Fig. R1.1 a trend very similar to the one of Fig. 3b of the original manuscript is observed despite the strong simplifying assumptions implied by Eq. (1) such as the neglect of the effects of wind shear, turbulent fluctuations and the radial dependence of the loads on the actuator disk. This detailed discussion is not included in the revised manuscript to keep the focus on the most important points but it is now mentioned that the increase of $C'_T$ with $\varphi$ is related to the combined effect of the tilt and of the associated decrease of the induction factor (revised manuscript, lines 147-148).

[Figure]

Figure R1.1 Dependence of $C_T'$ computed with Eq. (1) for each turbine of the upwind row using the $u_d(\varphi)$ data from the simulations for $\beta = 0^o$ and $\beta = -5^o$.

*- The author shows the dependency of $C_T'$ on different control parameters, but there is no mention of how e.g. the pitch angle affects the power coefficient $C_P$ (or $C_P'$ if you will). This should be clearly mentioned*

The dependence $C_P'(\beta)$ is shown in Figure R1.2 below for both tilt and yaw control. From this figure it can be verified that the relation $C_P' = \chi C_T'$ is a good approximation of the data (with $\chi = 0.9$).

This is now mentioned in the revised manuscript (lines 100-102).

[Figure]

Figure R1.2 Dependence $C_P'(\beta)$ of the rotor-based power coefficient on the pitch angle. Power coefficient predicted from the thrust coefficient as $\chi C_T'$ (with $\chi = 0.9$) are also reported for comparison. Panel $(a)$: tilt control. Panel $(b)$ yaw control

*The author frequently mentions achieving doubled or tripled power gains in high induction compared to baseline tilt/yaw control. Please be more specific in phrasing here to avoid confusion: mention explicitly the percentages, and the setup (e.g. Cossu 2020b has a three-row setup, achievable power gains are different than when looking at two rows as in the current study).*

This is done in the revised manuscript.

*Starting from line 131, the author discusses that he believes increasing thrust in tilted conditions should not impact turbine loading compared to standard operation, since the overall thrust force would not be higher than in the latter. However, Fleming et al (Renewable Energy 2014), have shown that tilt control can have a significant influence on blade bending and drivetrain torsion. Further increasing thrust could aggravate such issues. I believe that turbine loading could be an issue at higher induction scenarios such as considered here, and that conclusive statements warrant a detailed analysis using aero-elastic codes. This should be mentioned in the manuscript.*

This is right. Trying to modify the manuscript in this direction, however, I have realized that even with the suggested additions, the only partial discussion of turbine loading in tilted/yawed conditions remained at best only qualitative and at worst potentially misleading, so that I have completely removed this discussion in the revised manuscript where the need for further studies of the turbine structural loading is now clearly mentioned.

*Line 59: typo oveinductive should be overinductive*

Right. This is corrected in the revised manuscript.

*Line 82, the formula for $C'_T$ contains a $\pi$ which shouldnt be there*

Right. This is corrected in the revised manuscript.

*The appendix states use of a Schumann BC at the wall. What is the roughness length imposed at the bottom and, more importantly, what is the resulting turbulence intensity at turbine height? This tends to have a significant impact on power deficits and hence achievable gains.*

I thank you for this comment which has led me to find inconsistencies in the simulations input files. The used $z_0 = 0.01$ and the associated turbulence intensity (5.7% at hub height) of the incoming flow are mentioned in the revised manuscript (lines 109-110).

*A 3x3 km periodic precursor domain is probably too small to generate fully realistic turbulent flow structures. Does the author expect this to affect results in any way?*

Intuitively, I do not expect that running simulations in longer/wider domains would significantly affect the results, especially for the considered configuration, where the spanwise spacing of the turbines are smaller than those of natural large- (LSM) and very large scale motions (VLSM) of the boundary layer. In this case, indeed, a possible locking of the LSM and VLSM to the computational box should not have a significant influence on the global power gain. Furthermore, the fact that statistics are accumulated for more than one and a half hours (6000s) ensures their acceptable convergence as can be seen in all figures where data for each of the upwind-row turbines are shown and where it can be observed that the spread of data among turbines is small.

*The author rightfully mentions surprisingly little research efforts into combining yaw and induction control. An additional study that could be mentioned here is Munters, Meyers, 2018, Optimal dynamic induction and yaw control of wind farms: effects of turbine spacing and layout. J Phys Conf. Ser 1037, 032015, which investigates combined dynamic yaw and overinduction for a series of different wind-farm layouts*

This is right. This paper is now cited in the revised manuscript (lines 58, 210, 359-360).
————

I hope to have clarified the main issues raised in the report. I thank again Prof. Munters for his remarks and suggestions which have helped to improve the manuscript.

**Comments on the review of "Wake redirection at higher axial induction" - Reviewer 2**

Carlo Cossu

Laboratoire d'Hydrodynamique Énergetique et Environnement Atmosphèrique (LHEEA)

CNRS - Centrale Nantes, 1 rue de la Noë 44300 Nantes, France

January 13, 2021

I thank Reviewer 2 for his/her comments and suggestions and I appreciate the rapidity of the refereeing process. Each raised issue is addressed in detail below.

The reply process has been longer than expected because during the revision, in relation to the performed additional simulations and to a question raised by Referee 1 , I became aware that most turbine simulations input files were bugged leading to the inconsistent use of roughness lengths $z_0 = 0.15$ that were larger that used in the precursor simulation. I therefore had to rerun all the simulations with the correct value $z_0 = 0.001$, postprocess the results and redraw all the figures.

The manuscript has undergone a major revision, the main modifications being the following ones:

- All the presented results have been updated with the new simulations where the bug on the $z_0$ value was fixed. These corrected results, reported in the revised manuscript, are mostly similar to the previous ones so that the main conclusions of the manuscript do not change. Changes resulting from these new simulations are updated in the revised manuscript and are discussed, when appropriate, below.

- Additional simulation have been performed to analyze the role of the used turbine model, and in particular the effect of including wake rotation effects. The new results are presented and discussed in the new Appendix B and mentioned in the manuscript when appropriate.

- New figures have been added showing the mean streamwise vorticity and velocity fields in the cross-stream planes to highlight the role of the counter-rotating streamwise vortices forced by the tilt or yaw misalignement and discuss the role of wake rotation. A scheme has been added to define the tilt and yaw angles $\varphi$ and $\gamma$.

- The abstract and conclusions have been modified to make them more clear following reviewers' suggestions.

————

*1. The abstract mainly provides general introduction and the past research of the author. Please discuss the findings of the current research. You can quantify and further discuss the power gains due to tilted rotor and yaw control.*

I agree. The abstract has been largely rewritten accordingly.

*2. It is not clear why you have considered just two rows of turbines. I do not see any technical challenge in simulating for wind farms with more rows.*

I agree that the two-rows configuration is highly idealized and, as such, it is not representative of a typical wind farm and that technically I could have considered more turbine rows. However, the problem of considering many rows is that the results depend on the specific tilt/yaw angles enforced in each row. Thus, if I had considered more than two rows, the effect of increasing the induction in tilted/yawed turbines (which is the main message of this paper) would have been blurred by considerations/analyses of the

optimal combinations of tilt/yaw angles to be enforced in each row. I have therefore chosen to consider only two rows for which the results in term of $\beta$ and (a single value of) $\gamma$ or $\varphi$ remain relatively easy to interpret. The chosen configuration is indeed similar to the two-turbines case considered by Fleming et al. (2015) except for the fact that spanwise periodic distributions of turbines (the two rows) are considered instead of only two turbines. This was explained in the original manuscript (lines 61 to 64).

*But in case you think two-row wind farm simulation is sufficient, please discuss how you can link your findings to larger wind farms.*

Actually, I do not think that two-rows simulations are sufficient but that they are necessary. Indeed, in addition to the considerations discussed in point n.2, one should also consider that the computation of the optimal overinductive tilt or yaw control for realistic turbine arrays, where the optimal combination of tilt, yaw and pitch angles of all turbines has to be computed for a large number of wind directions and intensities, would be too computationally demanding if performed by means of large eddy simulations. This type of analysis is customarily based on less computationally demanding simplified sets of equations where the accurate modeling of the controlled wakes is of primary importance. In this context, the results presented in the present study should be used to improve/validate the existing simplified wake models in moderate to high-tilt/yaw and pitch angles regimes, particularly in the case of significant overinduction. Indeed, simplified models which are unable to reproduce the power gains results for the set of $\varphi, \gamma, \beta$ combinations and the idealized two-row array considered here would probably be unfit to predict annual power gains in more realistic settings.

This said, I agree, however, that the link between the studied idealized array and realistic configurations was not clear enough in the idealized manuscript. I have therefore summarized these points in the conclusions of the revised manuscript (lines 228-237).

*3. Line 81 to 83: Please add more explanation to clarify the relation between $\beta$ and $C_T'$. You can use blade element momentum (BEM) theory in order to describe the relations between $\beta$, lift and drag coefficients and the thrust coefficient.*

Additional explanations have been added to Appendix A (lines 265-270).

*4. Line 84 to 91: Add LES and other relevant equations.*

As in this study I have used the standard SOWFA code without changing its formulation, I prefer to refer the reader to the original papers to keep the focus on the main scopes of the study. The same is done in the many related studies based on SOWFA such as the cited ones of Fleming et al. (Renew. En. 2014) and Fleming et al. (Wind En. 2015) who also refrain from reproducing all the details of the formulation and refer to Churchfield et al. for the full details on the used formulation, including the LES equations. However, I agree, that section 2 lacked even some of the most basic information. This is fixed in the revised manuscript where more details on the used model (filtered Navier-Stokes equations with Boussinesq approximation) are now mentioned in section 2, lines 88-92 (they were briefly mentioned only in Appendix A in the original manuscript).

*Adding a schematic for the computation domain will be helpful too.*

The velocity fields reported in Figs. 1 and 4 of the original manuscript (Figs. 2 and 6 of the revised manuscript) show the full computational domain. This is now mentioned in the revised manuscript (line 118 and near the end of the caption of Fig.2).

*5. This may be beyond the scope of this manuscript, but how practical do you think it is to tilt blade by $\varphi = -30^o$? Higher tilt angle will significantly increase the flapwise bending and reduce the blade lifetime. You have mentioned about gravity load in line 137, but that is not very clear.*

I agree that the mention of loads, and gravity loads in particular was unclear in the original manuscript. I have removed it from the revised manuscript to avoid a potentially misleading only partial discussion of the structural turbine loading.

I also completely agree that the issue of the practicality of tilting turbines is important and deserves further investigations that are beyond the scope of this paper. However, let me note that given that positive tilt

is not immediately implementable in most of the installed horizontal axis wind turbines with upwind-directed rotor, possible drawbacks of the tilt on blade loads could be addressed in the design phase of a new generation of turbines with downwind-oriented rotors and highly flexible blades such as those discussed by Loth et al. (Downwind pre-aligned rotors for extreme-scale wind turbines, Wind Energy, 20, 12411259, https://doi.org/10.1002/we.2092, 2017).

That additional studies should consider the influence of overinduction combined to tilt and yaw on loads and the full aeroelastic response of the blades is now mentioned in the revised manuscript (lines 222-227).

*6. Line 116 to 122 and Figure 3(a): I do not understand why increasing $\beta$ (making it more negative) increases the power from the first turbine row. Wind turbines are usually optimized for the pitch angle around $0^o$. If that is the case with your turbine too, power output should be lower for $\beta < 0^o$. Increased thrust coefficients -for negative blade pitch angles- are simply caused by increased drag coefficients, and they will not necessarily translate into the higher power output.*

This is an interesting point [you are probably referring to Figure 2(b)]. I agree that it seems strange that more power can be produced by first-row turbines for the suboptimal values $\beta < 0^o$. There are however two reasons that can explain this apparently counterintuitive result:

(a) For the NREL5 turbine $\beta = 0^o$ corresponds, by design, to the maximum $C_P$ (at the optimal wind-tip speed ratio) but only for the reference case $\gamma = 0^o$, $\varphi = -5^o$ for which the optimization was performed. However, the data in Figure 2(b) do not pertain to reference values but to $\varphi = 30^o$ for which there is no guarantee that the maximum $C_P$ is obtained for $\beta = 0^o$.

(b) For the case of a (single) row of closely spaced (non-tilted/non-yawed turbines) Strickland & Stevens (Effect of thrust coefficient on the flow blockage effects in closely-spaced spanwise-infinite turbine arrays, J. Phys. Conf. Ser. 1618, 2020, doi:10.1088/1742-6596/1618/6/062069) show that "the power production of turbines in the row increases approximately linearly with $C'_T$ when compared to the production of a free-standing turbine". It is therefore possible that also in the present case the slight blockage effect of the first row increases when $C'_T$ is increased leading to an increase in the power production.

A note mentioning this has been added to the revised manuscript (bottom of page 7).

*7. You have not discussed how the tilt control and the yaw control influence the flow fields inside the wind farm. How do turbulence fields and shear stresses change as a result of those controls should be presented.*

I do not completely agree that I have not discussed how the tilt control and the yaw control influence the flow fields inside the wind farm because this is precisely what was done in Figs. 1 and 4 and the related discussion. Additional flow fields and discussion have, however, been added to the revised manuscript where the mean streamwise vorticity and velocity fields are now shown in crossflow planes in Figs. 3, 7 and B1 in order to better discuss the role of wake rotation.

I have shown the mean streamwise velocity fields because they are the ones which influence the mean power output which is the main subject of this study. I do not show the turbulence fields because they are mainly relevant for the analysis of the power and load *fluctuations*, an analysis which goes beyond the scope of the present study. However, that additional studies of load fluctuations for the presented overinductive tilt and yaw control is now mentioned in the revised manuscript (lines 222-227).

*Minor comments and corrections: 1. Line 10: an high → a high.*

Right. This is fixed in the revised manuscript.

*2. Line 8: of the produced → of that produced*

Right. This is corrected in the revised manuscript.

*3. Line 110: Is it $\varphi = -30^o$? You can add a schematic describing positive and negative directions for yaw, tilt and pitch angles.*

Actually, it is a positive tilt angle $\varphi = +30^o$ (see e.g. the discussion of Fleming et al. 2015 who write "With a positive tilt angle, the rotor would face downward, and for conventional upwind turbine designs, this would cause the blades to hit the tower").

A schematic describing positive and negative directions for yaw and tilt has been added in an additional figure (Fig. 1 of the revised manuscript). The schematic for the rotor collective pitch angle has not been added because it would have required a long discussion to avoid misunderstandings (indeed in a plot one should also discuss the local twist angle, the aerodynamic angle of attack and its definition, etc.). These angles are now mentioned in Appendix A (lines 267-270).

————

I hope to have clarified the main issues raised in the report. I thank again Reviewer 2 for his/her many remarks and suggestions which have helped to improve the manuscript.

[revised manuscript text omitted]